# Multi-Omics Reveal Interplay between Circadian Dysfunction and Type2 Diabetes

**DOI:** 10.3390/biology12020301

**Published:** 2023-02-14

**Authors:** Ashutosh Tiwari, Priya Rathor, Prabodh Kumar Trivedi, Ratnasekhar Ch

**Affiliations:** 1Metabolomics Lab, CSIR-Central Institute of Medicinal & Aromatic Plants (CIMAP), Lucknow 226015, India; 2Department of Biotechnology, CSIR-Central Institute of Medicinal & Aromatic Plants (CIMAP), Lucknow 226015, India; 3Academy of Council of Scientific and Industrial Research (ACSIR), Gaziabad 201002, India; 4School of Biological Sciences, Queen’s University Belfast, Belfast BT9 5DL, UK

**Keywords:** type2 diabetes, circadian dysfunction, transcriptomics, proteomics, metabolomics

## Abstract

**Simple Summary:**

Type 2 diabetes (T2D), a metabolic disorder, characterized by dysregulated glucose metabolism. Circadian rhythms, nearly 24-h biological oscillations, control daily biological functions including glucose metabolism, that are essential for survival. Circadian arrythmia caused by irregular meal timing, and sleep loss alters glucose metabolism and insulin production can result in metabolic condition, T2D. Understanding dysregulated circadian metabolism using systems biology approaches may provide solutions to treat T2D. Using multi-omics approach, present work correlates how circadian arrythmia caused by T2D alters different genes, proteins and metabolites.

**Abstract:**

Type 2 diabetes is one of the leading threats to human health in the 21st century. It is a metabolic disorder characterized by a dysregulated glucose metabolism resulting from impaired insulin secretion or insulin resistance. More recently, accumulated epidemiological and animal model studies have confirmed that circadian dysfunction caused by shift work, late meal timing, and sleep loss leads to type 2 diabetes. Circadian rhythms, 24-h endogenous biological oscillations, are a fundamental feature of nearly all organisms and control many physiological and cellular functions. In mammals, light synchronizes brain clocks and feeding is a main stimulus that synchronizes the peripheral clocks in metabolic tissues, such as liver, pancreas, muscles, and adipose tissues. Circadian arrhythmia causes the loss of synchrony of the clocks of these metabolic tissues and leads to an impaired pancreas β-cell metabolism coupled with altered insulin secretion. In addition to these, gut microbes and circadian rhythms are intertwined via metabolic regulation. Omics approaches play a significant role in unraveling how a disrupted circadian metabolism causes type 2 diabetes. In the present review, we emphasize the discoveries of several genes, proteins, and metabolites that contribute to the emergence of type 2 diabetes mellitus (T2D). The implications of these discoveries for comprehending the circadian clock network in T2D may lead to new therapeutic solutions.

## 1. Introduction

Circadian rhythms (in Latin, circa “approximate”; dies: “day”) are nearly 24-h biological oscillations that keep internal physiology in synchronization with the external environment. These biological oscillations are ubiquitously found in nearly all organisms including mammals [1]. The circadian system of mammals comprises a central clock situated in the suprachiasmatic nucleus (SCN) of the hypothalamus and peripheral clocks distributed throughout the body [2]. The master clock in the brain communicates with the peripheral clocks, including the liver, heart, adipose tissue, and pancreas, via synaptic and diffusible signals that regulate daily cycles of behavioral and physiological processes (Figure 1) [3]. In addition to these, gut microbiota also regulates the physiological processes via the activation of the central and peripheral clocks. The gut microbiome and circadian clock are intertwined, and the dysbiosis of the microbiome can dysregulate the circadian clock and vice versa. Dysregulation in the clock of gut microbiota leads to metabolic disorders such as T2D. In mammals, the master clock in the brain and peripheral clocks in tissues communicate with each other to establish a hierarchically structured circadian system that is essential for adaptability and survival [1]. The mammalian circadian clock is controlled by important factors including light, food, and temperature. The suprachiasmatic nucleus (SCN), which receives photic signals from light, synchronizes the peripheral clocks. Non-photic inputs like food synchronize peripheral clocks by communicating with the master clock in the brain [2]. There are three ways in which the SCN and bodily organs can communicate with one another: hormone secretion, the parasympathetic nervous system, and the sympathetic autonomous nervous system [4]. All cellular and tissue clocks must be synchronized to maintain homeostasis, and their desynchronization can alter physiological and behavioral processes, leading to metabolic disorders like obesity and T2D. A recent study found that the development of T2D is tightly coupled with circadian arrhythmia that dysregulate hepatic insulin sensitivity and energy metabolism [5]. Dramatic lifestyle changes, such as chronic sleep disorders in shift workers and irregular eating patterns in response to the industrialization of modern society, are connected to the increased prevalence of T2D.

### 1.1. Molecular Mechanism of Circadian Clock

#### 1.1.1. Transcriptional and Translational Feedback Loop (TTFL)

The mammalian circadian clock is regulated by a feedback loop system that consists of positively regulated clock genes, including *CLOCK*(circadian locomotor output cycles kaput), NPAS2 (neuronal PAS domain protein 2), and *BMAL1* (brain and muscle antlike protein-1), and negatively regulated clock genes, including *PERIOD (PER1, PER2,* and *PER3)* and CRYPTOCHROME (*CRY1/CRY2*) [2,6]. The positive loops produced by bHLH-PAS (basic helix-loop-helix; Per-Arnt-Single-minded) proteins are encoded by *CLOCK* and *BMAL1* [7]. These positively regulated clock genes (*CLOCK*&*BMAL1*) form a heterodimer complex (*CLOCK: BMAL1*) that binds to the E-box (5′-CACGTG-3′) and E′-box (5′-CACGTT-3′) in the promoters of target genes, which subsequently activate the transcription of target genes such as *PER* (*PER1* and *PER2*) and *CRY* (*CRY1* and *CRY2*) [8]. The cytoplasmic *PER* and *CRY* proteins are dimerized and translocate into the nucleus by the conclusion of the circadian day (CT12). PER-CRY complexes are accumulated in the nucleus and start to repress their expression [3]. The degradation of the negative loop proteins *PER* and *CRY* is required to maintain the clock rhythmicity and to initiate a new clock cycle of transcription once the suppression phase of TTFL has concluded [8]. Different signaling proteins are involved in the degradation. For instance, casein kinase, CK1, catalyzes the phosphorylation of the PER protein and mediates the targets for ubiquitination by βTrCP and destruction by the 26S proteasome [9]. There are AMPK and GSK-3β catalysts of the phosphorylation of *CRY1* and *CRY2* of the *CRY* protein, respectively [10]. FBXL3 (F-box and leucine-rich repeat protein 3) is one of the four subunits of SCFs (SKP1-cullin-F-box) that make up the protein ligase complex. It causes the ubiquitination of the phosphorylated *CRY* protein, which is intended for proteasomal destruction [11] [Figure 2 shows the mammalian molecular clock system]. The second feedback loop comprises of orphan nuclear receptor genes, including *ROR, REV-ERBα (NR1D1)*, and *REV-ERBβ (NR1D2)*, which stabilize the clock rhythmicity. The transcription of these genes is triggered by the binding of the heterodimer complex *CLOCK: BMAL1* to the promoter E-box [12]. The retinoic-acid-related orphan receptor response element (RORE), which is situated inside the promoter of Bmal1, is bound by *ROR* and REV-ERB proteins in a competitive manner after translation. Because of this, *ROR* proteins stimulate Bmal1 transcription, whereas REV-ERBa proteins inhibit it [13,14].

In addition to these layers of clock regulation, epigenetic modifications also regulate the clock rhythmicity [15]. Histone modifications are a crucial component of epigenetic modification that controls the shape of chromatin and the expression of genes. Histone modifications (including acetylation, methylation, phosphorylation, and ubiquitination) are crucial to the molecular mechanism of the circadian clock. For instance, the acetylation and deacetylation of H3 and H4 at the clock target gene are necessary for clock rhythmicity [16]. The clock protein possesses histone acetyl transferase activity (HAT), which is required for circadian regulation and for restoring the rhythmicity and activation of the clock gene in clock mutant cells [17]. Deacetylation selects the SIN3-HDAC (SIN3-histone deacetylase) complex that binds to the heterodimer complex (*CLOCK: BMAL1*)-bound DNA to inhibit *PER* transcription [18]. Coenzyme NAD+ (nicotinamide adenine dinucleotide) is an important circadian metabolite that is required for SIRT1 activity in order to deacetylate Histone H3 at the target clock gene. SIRT1, a deacetylase from the Sirtuin family, is necessary for the strong circadian expression of clock genes, such as *Bmal1, Ror*, *Per2*, and *Cry1* [19].

#### 1.1.2. Non-Transcriptional Circadian Rhythm in Mammals

Recent research findings have identified that post-translational and non-transcriptional mechanisms of circadian rhythmicity also exist in addition to the TTFL mechanism [8]. One of such critical findings is the redox oscillations of peroxiredoxin (PRDX) proteins in human red blood cells that play a critical role in maintaining and combating daily redox stress.

PRDX, an antioxidant enzyme involved in the reduction of hydrogen peroxide (H_2_O_2_), regulates intracellular peroxide levels through oxidation and reduction cycles to maintain the circadian rhythm. Peroxiredoxin belongs to one of the two groups: one-Cys peroxiredoxin and two-Cys peroxiredoxin. There are five peroxiredoxin (PRDX1-5) in group two-Cys peroxiredoxin and one peroxiredoxin (PRDX6) in group one-Cys peroxiredoxin found in mammals. One-Cys peroxiredoxin, such as those in the human PRDX6, undergo oxidation at the catalytic cysteine residue, resulting in the formation of a sulfenic acid cysteine (Cys-SOH). Cysteine joins with sulphur to produce a disulphide bridge in two-Cys peroxiredoxin, which is intramolecular in atypical variations and intermolecular in normal peroxiredoxin [20]. Glucose (an instantaneous source of energy) is also required for the clock rhythmicity and maintains cellular metabolism. Human RBCs (which lack nuclei) display the redox rhythm by the regulation of glucose metabolism, which maintains the daily clockrhythmicity in mammals [21]. In addition to the redox oscillations of PRDX, membrane electrophysiology is under circadian control in human red blood cells as it allows them to control K+ transport in a dynamic way, which in turn controls the rhythmicity of the clock in membrane electrophysiology [22]. Furthermore, cyclic adenosine 3’,5′-monophosphate (cAMP) signaling process are significant components of the oscillatory network. Cyclic adenosine monophosphate (cAMP) is a secondary messenger molecule that plays a fundamental role in cellular activities. The evolution of transcriptional rhythms is maintained by the daily activation of cAMP signaling, which is triggered by the transcriptional oscillator. Rhythmic cAMP signaling maintains the suprachiasmatic nucleus transcriptional loop and establishes the canonical pacemaker period, phase, and amplitude [23].

**Figure 2 biology-12-00301-f002:**
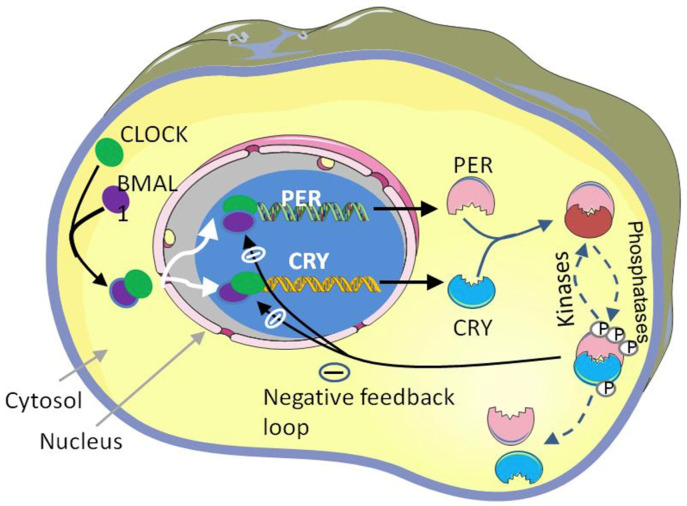
TTFL-based molecular mechanism of the circadian clock: The heterodimer complex *BMAL1: CLOCK* binds at the E-box of target genes and initiates the transcription. The cytoplasmic *PER* and *CRY* proteins are dimerized, translocate into the nucleus, and repress their gene transcription. The degradation of the negative loop proteins *PER* and *CRY* is required for the clock rhythmicity and to initiate a new clock cycle of transcription once the suppression phase of TTFL has ended (figure source: Powerpoint & Adobe illustrator).

## 2. Multi-Omics Approaches to Understanding Circadian Dysfunction in Various Tissues in Type 2 Diabetes

Mammalian cells are highly dependent on ATP levels for their critical functions, which are directly linked to glucose metabolism. Irregular glucose metabolism caused by insulin resistance is a major factor in T2D. Insulin is a peptide hormone that binds plasma membrane-bound receptors in cells to coordinate an integrated response to nutrient availability [24]. This crucial hormone is produced by the pancreas, which helps the utilization of blood glucose in metabolic tissues, including muscle, fat, and liver, where it is used for energy production. These metabolic tissues perform specific roles in metabolic homeostasis, demanding tissue-specific insulin signal transduction pathways. In addition to these metabolic tissues, the brain uses 50% of the glucose homeostasis of the body and indirectly connects to insulin secretion. The brain-centered glucoregulatory system can lower blood glucose levels via insulin-dependent mechanisms [25].

All of these tissues possess internal biological clocks and play a crucial role in regulating glucose homeostasis and energy production. Misalignment of endogenous cellular biological clocks (circadian disruption) in these tissues and fasting-feeding cycles lead to impaired glucose homeostasis, which causes T2D. The internal timing of each tissue plays a critical role in regulating glucose homeostasis [26]. Omics play an important role in understanding circadian arrhythmia in T2D. Especially, omics trilogy, genomics, proteomics, and metabolomics may provide information related to circadian dysregulated genes, proteins, and metabolites in different tissues associated with T2D.

Genomics is the study of the genetic or epigenetic sequence information of organisms. Importantly, circadian clock genes were among the first genes identified that control behavior. Starting from the identification of the first circadian mutant period in fruit flies, a forward genetic screen was implemented in mammals and mice and it identified the first mammalian circadian gene clock. Thereafter, research findings added many additional genes to the core clock loop. A total of 43% of the human genome exhibits circadian rhythmicity [27]. In addition to genomics, proteomics (the analysis of a set of proteins found in cells, tissues, and organisms under certain, predetermined conditions using high-throughput analytical platforms) may provide information related to circadian-dysregulated proteins in T2D systems. It has significant promise for understanding human physiology, protein biomarkers, and medicine that is associated to health and illness [28]. The posttranslational modification (PTM) of proteins is characterized by the covalent processing events of amino acids side chains by the addition of some groups, such as acetyl, glucosyl, methyl, and phosphoryl. PTMs have an impact on almost every aspect of clock biology; in certain situations, they are crucial for clock operation, while in others, they offer additional levels of regulatory fine-tuning [29]. The most prevalent PTM is protein phosphorylation, which involves adding a phosphate group to serine, threonine, or tyrosine. It is widely reported that the core clock protein has been shown to be crucial for the generation of circadian rhythms [30]. Further, the downstream of omics, metabolomics (the study of small molecule metabolites within the cells, tissues, and biological fluids at a specific time and with a specific physiological condition using high throughput analytical platforms, i.e., NMR/LC-MS/GC-MS/CE-MS) may provide information related to circadian-dysregulated metabolites and metabolic pathways. The rhythmicity of metabolites is essential for physiological functions. Nearly 25% of the human blood metabolome is clock-regulated [31], and T2D affects the daily rhythmicity and concentration of circulating metabolites across 24 h. [32]. Investigation of circadian-dysregulated genes, proteins, and metabolites in various tissues of a T2D system may help to understand how a dysregulated clock system causes desynchrony in metabolic tissues that leads to T2D.

### 2.1. Master Circadian Clock; Suprachiasmatic Nucleus (SCN)

The SCN is a master clock that resides in the brain and coordinates the circadian oscillators, primarily by maintaining behavioral cycles of activity/rest and feeding/fasting with the aid of synaptic and diffusible components through neuroendocrine and neuronal outputs [33,34,35]. The SCN consists of a dorsomedial shell and a ventrolateral core with approximately 20,000 neurons, and it has the potential to generate clock rhythmicity [36]. Robust circadian rhythms of clock genes *Per1, Per2*, and *Bmal1* were found in mice SCN. Another important study conducted on rats identified that the SCN has the capability to regulate the glucose concentration in plasma over the course of 24 h. [37] and that it may also have a role in glucose tolerance and insulin sensitivity [38]. The intrinsically photoreceptive retinal ganglion cell (ipRGC) has the capability to processes the photic signals from the retina, which activates the SCN. The ipRGC depolarizes photoreceptors along the RHT by glutamate and pituitary adenylyl cyclase-activating peptide (PACAP) [36,39,40]. The rhythmicity of SCN core neurons is controlled by photic signals through a variety of neurotransmitters, including vasoactive intestinal polypeptide (VIP), gastrin-releasing peptide (GRP), and SP [41]. In humans, the arrhythmicity of the SCN is linked to the onset of metabolic diseases including T2D, and can cause the peripheral clocks to become uncoupled from the central pacemaker [42,43]. The SCN activates the release of various hormones, including the corticoliberin/corticotrophin-releasing hormone (CRH) and neuropeptide hormone from the paraventricular nucleus of the hypothalamus, and stimulates the rhythmic release of the adrenocorticotropic hormone (ACTH) from the anterior pituitary gland. The ACTH regulates the release of glucocorticoids (GC; cortisol), which have a role in a number of biological processes, including gluconeogenesis, lipid metabolism, inflammation, and immunological activities [44]. Photic signals can alter the glucocorticoid (GC) levels in plasma by stimulating the sympathetic nervous system. According to a study, non-lesioned animals had considerably higher levels of glucocorticoids (GC) in their plasma than animals with a lesioned SCN. Photic signal-induced gene expression and cortisol production are tightly regulated by the circadian clock [45]. In addition to glucocorticoids, the SCN controls the rhythmicity of melatonin. Melatonin, a hormone with antioxidant and anti-inflammatory properties, has been linked to diabetes mellitus associated with circadian arrhythmia. T2D is thought to develop as a result of decreased melatonin levels and a functional relationship between melatonin and insulin [46].

### 2.2. Gastrointestinal Track (GI Tract) and Microbiome

As the body first point of contact with ingested nutrients, the gastrointestinal (GI) tract is an essential system for maintaining metabolic homeostasis [47]. The GI tract is essential for maintaining proper digestion and nutrient absorption. Nutritional absorption rates are influenced by gut motility, which has been shown to be controlled by the circadian clock [48]. Protein from food is fragmented into amino acids, dipeptides, and tripeptide under the influence of digestive enzymes. These components are moved over the intestinal barrier via the high-capacity, low-affinity peptide transporter 1 (PEPT1). Since PEPT1 is a proton-dependent co-transporter and is dependent on sodium–proton exchangers (NHE3), which are known to be clock-controlled genes with putative E-boxes in their promoters proven to be regulated by Clock/Bmal1 and significant in the rhythmic physiology of the GI tract [48,49]. Additionally, microbes (i.e., the gut microbiota) within the GI tract are engaged in a number of processes, such as the production of micronutrients and the metabolism of drugs. The human digestive system contains about 100 trillion microbes. Numerous aspects of human health, including innate immunity, hunger, and energy metabolism, are influenced by gut bacteria. The clock rhythmicity of gut microbiota and the GI tract is crucial for physiological processes [50]. There are several factors, including medications, antibiotics, food additives, and pesticides, that can have a negative impact on the clock of gut microbiota. The perturbation of the clock rhythmicity of the GI tract and gut microbiota is contribute to the development of metabolic diseases such as T2D and may be a risk factor for a number of gastrointestinal diseases [51]. Circadian rhythms are crucial in maintaining microbiome composition [52]. The clock rhythmicity of the microbiome influences the daily rhythmicity of histone acetylation in intestinal epithelial cells, which regulates metabolic responses [53]. The maintenance of metabolic homeostasis is aided by circadian processes of microbiota–host interactions. There is a functional relationship between the diurnal oscillation of gut microbiota and metabolic balance, as shown by the discovery of clock-regulated metabolic pathways associated with xenobiotics, branched-chain amino acids, fatty acids, and taurine metabolism. The loss of diurnal oscillation in the gut microbiome composition and associated rhythmic mechanisms can contribute to the emergence of metabolic disorders, T2D [54].

The circadian rhythmicity of the gut microbiome is crucial for the regulation of the host metabolism. The diurnal rhythmicity of the gut microbiota is important for proper functioning, and diurnal rhythmicity is influenced by several variables, including food composition, the host circadian rhythm, and the time of feeding. One of the significant studies reveals that T2D leads to a loss of diurnal rhythmicity in gut microbiota that alters the various circulating metabolites. There are multiple metabolic pathways, such as histidine, betaine, and the methionine pathway, that have shown altered diurnal patterns in T2D. There is several gut microbiotas (such as Bifidobacterium, Clostridium, Bacteroides, Prevotella, and Alercreutzia) involved in the histidine metabolic pathway that has also been altered by diabetes [55].

### 2.3. Liver

It is the largest internal organ found only in vertebrates that helps to control postprandial glucose metabolism [56]. Hepatic glucose production rates are under circadian control and have a significant impact on the daily regulation of insulin sensitivity and meal tolerance. Approximately 10% of genes exhibit circadian patterns of expression in liver tissues [57]. At the cellular level, hepatocytes exhibit circadian expression patterns for metabolic genes that are critical for controlling hepatic glucose absorption and storage (such as Glut2 and Gck), as well as for hepatic glucose production (e.g., Pepck and Gcgr) [58]. The deletion of the core clock gene, Bmal1, in mice liver results in the loss of the circadian regulation of genes and of Glut2—which is important for controlling hepatic glucose metabolism—the loss of the latter causing dysregulation in systemic glycemia and glucose tolerance. Thus, the liver-specific deletion of Bmal1 mimics the hepatic pathophysiology in T2D, which is characterized by elevated oxidative stress, decreased insulin signaling, ectopic lipid buildup, and hepatic insulin resistance [59]. PER 2-mutated mice rapidly develop liver damage, demonstrating the close relationship between liver function and the circadian clock [60]. In addition to these, protein phosphorylation is the primary mechanism that controls the body internal clock, which is responsible for the regulation of both the metabolism and physiology of the body. A significant study conducted on mice liver revealed that the phosphorylation cycle of the *CRY1* protein at S588 residues peaked at CT0, while the phosphorylation cycle of the *CRY 2* proteins at S55 peaked at end of the active phase, i.e., CT21. Levels of *CLOCK* proteins stayed stable throughout the day, unlike other clock proteins and the phosphorylation cycles in S446A and S440/S441, which may serve as the mechanism which regulates rhythmicity [61]. Additionally, the circadian clock is an important modulator of protein acetylation through the regulation of nutrient metabolism. Lysine acetylation is a reversible PTM involved in the regulation of gene expression, subcellular localization, and enzyme activity. It is highly relevant in the aspects of circadian biology, and it has displayed subcellular localization-specific phases that are associated with metabolites in the regulated pathways. The circadian clock is a significant regulator of the deacetylase activity of SIRT3 that affects the function of mitochondria. In rhythmically acetylated proteins, mitochondrial proteins are overrepresented, and their deacetylation is strongly associated with SIRT3 activity that is dependent on the level of NAD+. A study revealed that global acetylation increased in Bmal1 KO mice but marginally reduced in *Cry1/2* DKO mice. This shift in global acetylation may thus be a result of the circadian clock control over NAD+ production and its subsequent control over SIRT3 activity. There are many proteins that are rhythmically acetylated, including those in ethanol metabolism, ketogenesis, and mitochondrial oxidative phosphorylation. Many proteins in these pathways have shown dysregulated acetylation levels in *CRY 1* and *BMAL1* KO mice [61,62].

The liver possesses separate circadian clock processes that are specific to the NAD+ salvage pathway and glycogen turnover [63]. A significant downstream metabolomics study conducted on mice reported different metabolites in the context of overall energy balance and chronic nutritional stress (high-fat diet, HFD). Amino acids, i.e., branched-chain amino acid (BCAA) levels in the liver oscillated for 24 h on chow and HFD, its level increasing in the dark period and decreasing in the light period. A chronic HFD may increase local sensitivity to endogenous glucocorticoid action. Indeed, on HFD, blood glucose remained consistently raised, but in mice given chow, the oscillation of glucose in the liver was attenuated and increased only at a specific time point (ZT12). Liver glycerol and non-esterified fatty acids (NEFAs) were greatly elevated on a HFD, emphasizing the pathological accumulation of physiologically active lipids linked to insulin resistance (T2D) [32,64].

### 2.4. Skeletal Muscles

Skeletal muscle is one of the major tissues that make up around 45% of overall mass and regulate the physiological process. The timing of exercise and food intake, as well as light signals to the SCN, can have a direct impact on the circadian rhythms of skeletal muscles. Fasting is another factor that significantly reduces the myogenic regulatory factors MyoD and myozenin mRNA expression in skeletal muscle, which is greatly increased during dark periods [65]. In skeletal muscles, more than 2300 genes exhibit circadian patterns of expression that are involved in a variety of processes, such as myogenesis, transcription, and metabolism [66]. On a cellular level, primary human myotubes exhibit a circadian rhythm, and the metabolic disease statuses of donor groups are connected with the amplitude of the clock gene REV-ERB [67]. It is an insulin-sensitive metabolic tissue that uptakes around 80% of postprandial glucose. Skeletal-muscle-specific Bmal KO mice affected both tissue glucose metabolism and systemic glucose homeostasis [68]. The discrepancies in insulin hormone sensitivity are caused by the circadian misalignment of the skeletal muscles. One of the important transcriptomics studies revealed the expression of the molecular clock gene in skeletal muscles under controlled and dysregulated conditions. In controlled conditions, the expression of clock genes *BMAL1, CRY1*, and *PER2* showed diurnal differences, while in dysregulated conditions, *PER* and *CRY* were blunted. Additionally, the pathway analysis (in the form of gene set enrichment) of dysregulated clock conditions revealed the set of altered gene expressions that are associated to the metabolism of fatty acid and peroxisome proliferator-activated receptor (PPAR) signaling, which regulates lipid metabolism [69]. The gene enrichment analysis of skeletal muscles found that *PER3* and *BMAL1* showed differential rhythmicity and that the *CLOCK* gene was exclusively rhythmic in myotubes. One of the significant studies revealed that siRNA mediated depletion of gene, OPA1 which is associated with the maintenance of the membrane potential of mitochondria and disrupts the expression of molecular clock genes, such as *PER2, PER3*, and *NPAS2*, in primary myocytes. As a result, irregularities in the daily rhythm of mitochondrial activity may involve skeletal muscles’ insulin resistance in T2D [70]. In addition to these a balance between protein synthesis and breakdown is essential for the maintenance of muscle mass. Excessive protein degradation, which is predominantly triggered by the ubiquitin–proteasome system as a result of aging and a number of illnesses including diabetes and denervation, causes progressive muscle atrophy [66]. One of the important studies revealed that the degradation of muscle protein is often inferred by the level of peculiar amino acids such as3-methylhistidine, which is excreted in the urine [71]. The amino acid 3-methylhistidine was detected only in the serum and skeletal muscles, which showed 24-h circadian oscillations, and it dramatically increased on a HFD, indicating the breakdown of muscle protein. The physiological functions of muscles depend on the rhythmicity of the mitochondria, a vital organelle for eukaryotic cell maintenance that is engaged in metabolic processes, including the production of ATP through oxidative phosphorylation and the synthesis of lipids and amino acids. The oxidative phosphorylation and generation of ATP is tightly clock-controlled. These circadian oscillations depend on the clock modification of dynamin-related protein 1 (DRP1), a crucial modulator of mitochondrial fission [72,73]. According to a recent study, dysregulated rhythmic mitochondrial metabolism may be the root cause of irregular rhythmic cellular metabolism and molecular clock mechanisms in the primary myotubes of T2D patients [74].

Additionally, exercise is also an important factor that influences the daily activity of the body via the activation of the brain and peripheral clocks. A significant study was conducted on diabetic humans influenced by the timing of exercise. In this study, participants who finished all of the exercise sessions were taken into account for the omics analysis (n = 8). The experiment included three sessions per week of high-intensity interval training (HIT) for two weeks, either in the morning (08:00 for n = 5) or the afternoon (16:45 for n = 3), followed by a two-week washout period and two further weeks of HIT at the opposite times. Acyl-carnitines, which are thought of as a transport form of fatty acids, can be used in mitochondria for energy generation. Acyl-carnitines generate more reactive oxygen species than pyruvate does during mitochondrial oxidation, and this may contribute to insulin resistance (T2D) in skeletal muscles. Levels of acyl-carnitines in skeletal muscles were elevated by morning and afternoon HIT. The training-induced metabolic crosstalk among tissues depended heavily on acyl-carnitines. When compared after morning to afternoon HIT, skeletal muscle lipoprotein levels were greater, while mitochondrial complex III was less abundant. Morning and afternoon exercise is related to levels of antioxidant metabolites and acyl-carnitines in skeletal muscles [75].

### 2.5. Pancreas

The pancreas is a multifunctional organ that regulates metabolism and digestion through a multitude of endocrine and exocrine functions. It consists of pancreatic islets of Langerhans, which contain alpha, beta, gamma, and delta cells, and each cell has the ability to produce a variety of hormones [44,76]. It has the capability to produce the insulin hormone in a rhythmic fashion and maintain glucose homeostasis. There are significant transporters, such as glucose transporter type-2 (GLUT-2), which have a high capacity and a low affinity for glucose, are expressed by the islets of Langerhans’ beta cells, and are responsible for detecting variations in blood glucose levels [33]. Glucose is transported into beta cells via the GLUT-2 transporter and subsequently passes through the glycolytic pathway and mitochondrial metabolism, increasing the amount of cytoplasmic ATP and activating ATP-sensitive potassium channels (KATP). Upon the depolarization of the plasma membrane, voltage-dependent calcium channels (VDCC) allow the calcium ion to enter the cell. The increased intracellular calcium concentration is responsible for the fusion of the insulin granules with the cell membrane and for the subsequent release of insulin [33,34]. The biological clock influences hepatic glucose synthesis and uptake, as well as glucose tolerance, through regulating feeding-induced insulin responses [35]. The risk of the development of diabetes may be increased by a variety of factors, including sleep deprivation and circadian misalignment brought on by changing sleeping patterns [76]. Over the course of a day, the pancreas exhibits variable expressions of circadian clock genes, such as PER1, *PER2, BMAL1, CLOCK*, and REV-ERB ALPHA [48]. Furthermore, it was demonstrated that the insulin-secreting cells of the human pancreas, known as islets, had a high transcription factor of the d-binding protein (DBP) and a high thyrotrophic embryonic factor (TEF), as well as a large amount of clock-controlled output genes [77]. Insulin secretion from pancreatic islets is closely controlled by clock genes, and perturbation contributes to the onset of T2D.One of the important studies revealed that the amplitude of the clock genes PER2, PER3, and CRY2 decreased in the islets of a donor with T2D. The expression of PER2, PER3, and CRY2 mRNA levels positively correlated with insulin content [78]. A significant study conducted on intact islets and islet cells of T2D patients revealed that the attenuated molecular oscillators had reduced circadian amplitude and decreased synchronization capacity in vitro. The amplitude of circadian clock genes (*CLOCK, BMAL1, PER, CRY, Rev-erba*, and *DPP*) significantly decreased in T2D as compared to non-diabetic control (ND) islet cells, and levels of NFIL3 was slightly elevated, though BMAL1 and CRY1 remained the same [79]. There is a strong relationship between circadian rhythms and the development of metabolic disorders such as T2D. One of the important study found the reduced amplitude of the mRNA transcript of *BMAL1, PER1, PER2,* and *PER3* in leucocytes of T2D as compared to ND ones [80].

As one of the important metabolite classes, lipids are considered a crucial component of the cell membrane and participate in the signaling pathways by which cells communicate with one another and with their surroundings [81]. A recent study revealed that the circadian clock operating in human pancreatic islets is necessary for the temporal orchestration of lipid homeostasis and that the dysregulation of the circadian rhythm of the islet of lipid metabolism inT2D affects insulin production and membrane fluidity. The temporal profiles of human pancreatic islets (10 ND and 6 T2D donors) reported that among 329 lipid species across 8 major lipid classes, 5% exhibited circadian rhythmicity in ND human pancreatic beta islets synchronized in vitro. In the majority of the donors, a peak in rhythmic lipid accumulation was seen 12 to 16 h after in-vitro synchronization. Among the several kinds of lipids, Phosphatidylinositol (PI) lipids are particularly enriched in all donors including T2D and ND donors. Few phospholipids’ metabolites, particularly from the PE and PI lipid classes, were down-regulated in the T2D group. At 24 h, the simultaneous rise in PC levels and the fall in diacyl PE levels led to a marked rise in the PC/PE ratio, which is known to affect cellular calcium homeostasis and ER function [82].

### 2.6. Cardiovascular System

The cardiovascular system, which comprises the heart, arteries, veins, and capillaries, distributes adequate blood throughout the body. The cardiovascular processes, such as endothelial activity, thrombus formation, blood pressure, and heart rate, are governed by the circadian rhythm. Throughout the day, several cardiovascular processes change as a result of the clock-controlled oscillation of cellular activity [83]. Various pathophysiologcal conditions, such as atherosclerosis, insulin resistance (T2D), a dampening of blood pressure rhythmicity, and a reduced production of vasoactive hormones and neurotransmitters, are associated with cardiac arrhythmia or disrupted 24 h rhythms [84]. Studies have indicated that diabetic people have a two- to three-fold increased risk of having cardiovascular diseases (CVDs) than healthy individuals [85]. Diabetic patients frequently have greater resting heart rates and less variability in their heart rates throughout the day than non-diabetic patients do, which causes the unnecessary consumption of oxygen in the myocardium with a reduced blood supply [86]. T2D is a well-known risk factor for CVDs such as myocardial infarction [87], and its incidence and prognosis are all influenced by circadian rhythms [83]. The hallmark of acute myocardial infarction (AMI) is the permanent oxygen deprivation-induced damage to cardiac muscles. Numerous earlier studies identified that the circadian distribution of AMI started mostly in the morning [88,89]. It is plausible to assume that the morning rise in sympathetic activity, plasma cortisol and renin levels, blood pressure, and heart rate are the major causes of this peak in AMI [90]. Additionally, it has been shown that fibrinolytic activity has decreased and that platelet aggregability and coronary tone have increased [91]. Daios and their colleagues studied diabetes patients with circadian patterns of AMI and atrial fibrillation (AF) in a Mediterranean country [92]. AMI has a circadian rhythm that peaks around midnight (21:01–23:59) and is more common in individuals between the ages of 40 and 65. In relation to the AF initiation, the incidents were more common at midday (12:00–14:59). Numerous factors, including intensive anti-diabetic medication, glycemic fluctuation, and various stages of insulin secretion and resistance, may be responsible for such pattern [93]. The onset of AF may follow a circadian rhythm since insulin resistance is higher in diabetic individuals in the morning and evening but decreases after midday. As a result, after lunch, hypoglycemia could happen more frequently [92].

Approximately 8% of genes exhibit the circadian patterns of expression in the heart tissues [83]. A genomics investigation may be able to identify the circadian rhythmicity in the cardiovascular system associated with various patho-physiological processes. Myocardial infarction and stroke incidence vary during the day, which may reflect the molecular clock or the timing of environmental stress exposure. One significant study employed on mice and demonstrated the importance of the core clock gene *Bmal-1* in controlling blood pressure and heart rate [93]. In addition *to Bmal-1, Per-2* also maintains the cardiovascular processes in mammals. When the *Per2* gene is mutated, it causes the aortic endothelial cells to produce a reduced level of nitric oxide (NO) and vasodilatory prostaglandins and more cyclooxygenase-1-derived vasoconstrictor molecules (s) [94].

Another important study conducted on mice demonstrated the surprising effects of clock machinery on systemic glucose metabolism in the heart. Since heart function is known to be linked to glucose tolerance, a disturbance of the clock mechanism in the heart may potentially influence systemic glucose metabolism. The heart-specific deletion of the core clock gene, Bmal-1, in mice exhibited a reduced insulin-induced phosphorylation of Akt in the liver, thus indicating that the deletion of Bmal-1 in the heart causes hepatic insulin resistance [95]. In addition to these, variations in the circadian gene also influence cardiovascular processes. Single nucleotide polymorphisms (SNP) in clock genes are associated with metabolic disorders such as hypertension and T2D; these conditions increase the risk of AMI [96]. Of note is a study by Skrlec and their colleague that explored the possible association between single nucleotide polymorphisms in three circadian rhythm genes (ARNTL, CLOCK, and PER2) and myocardial infarction inT2D. They found evidence of an association between AMI patients with T2D and ARNTL gene variants rs12363415 and rs3789327 and the CLOCK gene variants rs6811520 and rs13124436 [87]. There are several clock genes dysregulated in heart tissue associated with various physiological effects (Table 1).

Proteomic study is nevertheless constrained by sensitivity. For instance, in a transcriptome analysis, several thousand genes are often identified as oscillatory, whereas only a few hundred may be found in a proteomic examination of the same tissues [97]. One of the important circadian proteomics studies of the heart conducted on mice revealed that 8% of the soluble cardiac proteome was oscillatory. Importantly, the authors showed that the soluble proteome in CCM mice hearts contained 4% of the proteins that were expressed differently from the wild type. These proteins include several important key enzymes (aspartate aminotransferase, dihydrolipoyllysine succinyltransferase of 2-oxoglutarate dehydrogenase, and mitochondrial pyruvate dehydrogenase (E1a)) that are crucial for the various metabolic pathways [98]. There is little evidence found on metabolomics in terms of the dysregulated circadian clock in the heart associated with T2D.

### 2.7. Renal System

The mammalian kidney comprises nephrons, functional units that are essential for maintaining bodily homeostasis since they are primarily responsible for the elimination of metabolic waste and the control of extracellular fluid volume, electrolyte balance, and acid-base balance. Aside from these, the kidneys are also in charge of producing a few hormones that control various metabolic processes [99,100]. These functions rely on an internal clock mechanism that was previously thought to be regulated by the central nervous system. One of the important events is that blood pressure (BP) typically displays a circadian rhythm in healthy individuals, with higher levels during their wake cycle and lower levels during their rest cycle. The diurnal variation in blood pressure results from the change in the functions of the kidney [101]. The filtration unit of nephron called renal corpuscle or glomerulus consists of a tortuous bundle of blood capillaries located within the Bowman capsule, which is responsible for the filtration of the blood [102]. The circadian system controls the glomerular operations. The glomerular filtration rate (GFR), which is determined by the clearance of inulin and creatinine, peaks in healthy persons between 2–3 p.m. during the day and falls in the middle of the night [103]. The circadian rhythm for effective renal plasma flow (ERPF), as determined by p-aminohippurate clearance, likewise peaks during the day time, however it does so later in the afternoon compared to the GFR [104]. Additionally, urinary albumin and ß2-microglobulin excretion exhibit diurnal fluctuation in a period that is comparable to that of the GFR in healthy persons [102]. One of the important studies conducted on (male) Wistar rats revealed that the glomerular capillaries express the core clock protein, Bmal1, and the clock output protein, D site albumin promoter binding protein (Dbp), and that these expression levels change at different times of the day [105]. The intrinsic property of kidney, glomerulotubular balance, participates in the co-transport and exchange of fluids within the nephron. One of the important transporters, sodium–hydrogen exchanger 3 (NHE3), helps to reabsorb Na^+^ [106]. The co-transporters sodium glucose co-transporter -2 (SGLT-2) and SGLT-1 are responsible for reabsorbing most of the filtered glucose along with Na^+^, and they can be disrupted in hyperglycemia to result in osmotic diuresis in diabetes [107,108]. The importance of circadian rhythms in this illness has gained attention in recent years. One significant study discovered that individuals with type 1 and type 2 diabetes who did not experience the usual circadian BP change were more likely to die than those who had a normal circadian variation in BP. Additional research has revealed that patients with either type 1 or type 2 diabetes might exhibit abnormal circadian rhythms in blood pressure prior to kidney impairment [109,110,111].

The transcriptional mechanisms of the circadian clock are regulating an increasing number of genes. Several clock-controlled genes have been identified in the kidney through gene expression profiling. One of the important clock genes, Per1 regulates the daily fluctuation of sodium excretion and BP. The rate-limiting subunit of the epithelial sodium channel is encoded by Scnn1a, whose basal and aldosterone-dependent transcription is aided by Per1. Per1-mutated mice have lower expressions of Scnn1a in the renal medulla and, as a result, excrete more urinary sodium than the wild type does [112]. There are several clock-controlled transport genes identified in the kidney (Table 2). More omics studies are needed to explore the circadian-dysregulated clock in the kidneys of patients with T2D.

### 2.8. Biological Fluids

Biological fluids (plasma, serum, saliva, synovial fluids, and cerebrospinal fluids) are quite complex and comprise of micro and macromolecules. Human plasma proteins are controlled throughout the course of a 24-h day by the endogenous circadian SCN clock, the behavioral cycles of sleep, wakefulness, and food intake, as well as by interactions between the circadian and behavioral cycles. Circadian misalignment affects the 24-h protein patterns and amplitudes, which may lead to metabolic disease, T2D. One study revealed that the timing of food intake and sleep alters the 24-h rhythmicity of the plasma proteome of humans. A total of 62 proteins out of 1129 proteins were analyzed in 24 h during clock misalignment versus alignment. A total of 38 proteins increased out of the 62 proteins that were associated with multiple pathways linked to the alpha subunit of PI3K signaling, which is involved in the insulin-signaling pathways. A total of 24 proteins decreased out of these 62 proteins associated, with multiple pathways linked to antigen presentation and processing and IFN (interferon) signaling. These altered proteins during circadian misalignment were associated with multiple biological processes [113]. Along with plasma proteins, a number of plasma metabolites were also shown to be relevant to clock dysregulation. One study was conducted on overweight/obese (OW/OB) T2D patients and quantified 130 metabolites in plasma every two hours over the course of a 24-h period to identify the dysregulated circadian metabolome. In comparison to T2D and OW/OB groups, 56 of 130 metabolites showed significant differences, with 34 of the 56 metabolites being amino acids, biogenic amines, acyl-carnitines and 2 sphingolipids (SM C20:2 and SM C26:1were significantly higher in concentration in T2D [69,81]. T2D creates microvascular complications such as diabetic retinopathy, which is characterized by affected blood vessels and retinal nerve tissue, which is indispensable for the visualizations. One circadian clock study was conducted on mice and collected blood plasma samples, which were collected at two time points, i.e., during the day and the night. A total of 79 of the 747 distinct metabolites found in plasma were significantly changed by diabetes and the time of the sample collection. No metabolites were discovered to be associated with the circadian pathway during the day, although histamine, L-histidine, L-cysteine, adenosine, sorbitol, and choline were. D-pantothenic acid, oleoglycerol, and circadian rhythm signaling were discovered to correlate with DR and cyclic AMP throughout the night [53].

The altered genes, proteins, and metabolites in T2D associated with circadian arrhythmicity are presented in Table 3.

## 3. Management of T2D through Circadian Intervention

T2D treatment through circadian rhythm intervention is indispensable for glucose homeostasis. There are therapeutic strategies targeting circadian rhythms for T2D treatment. These strategies include medication and non-medication [2]. Metformin is a medication used to treat T2D, and research has shown that it works by activating an enzyme called AMP-activated protein kinase (AMPK). Most clock and metabolic genes in the liver are in a phase as a result of metformin treatment. Because metformin increases AMPK activity, the negative feedback loop’s half-life is decreased and the activity of the positive loop of the circadian clock is increased [114].

Melatonin is a hormone produced by the pineal gland. Its production is strongly regulated by the master clock and suprachiasmatic nucleus (SCN), and it has a positive impact in treating T2D [46]. This hormone is associated with various physiological functions, most importantly the entrainment of signals to the SCN for regulating sleep, which is critical in the treatment of sleep and circadian rhythm disorders [115]. Circadian disturbances, such as too much artificial light at night, shift work, or night work, affect the cycles of melatonin and suppress its secretion, causing sleep deprivation and stress, which leads to the emergence of disorders such asT2DM. In rodent species, there is evidence that melatonin may help with insulin resistance and glucose balance [116]. Several studies have also demonstrated the beneficial effects of melatonin on T2D patients’ blood glucose management and sleep quality [117]. Treatment for T2DM may be aided by focusing on the circadian rhythm. The molecular clock was specifically addressed by the REV-ERB agonist SR9011, which was a potential option for treating obesity and improving glucose metabolism in T2DM animal models [118]. An important modulator, nobiletin, a natural polyethoxylated flavone, reduced peripheral lipid accumulation, improved glucose tolerance, and restored insulin sensitivity in the liver [119].

In addition to drug medication, light and food play a crucial role in the management of T2D. Since light signals are primarily received by the SCN, improving daily light exposure can improve circadian synchronization [120]. Mood disorders, depression, and circadian anomalies such as shift work and sleep difficulties have all been treated using bright light therapy (BLT). The circadian system was helped by high-intensity light exposure each morning, which reduced body weight and enhanced glucose tolerance or improved T2D in sand rats [121,122]. Further, there is mounting evidence that the chrono-nutrition strategy consists of calorie restriction, intermittent fasting, and time-restricted feeding [123,124], which have positive effects on metabolism that are a result of how they affect the circadian clock. Similarly, time-restricted feeding (TRF) and intermittent fasting (IF) also reduce metabolic abnormalities by re-establishing one circadian clock, which also involves the gut microbiota [125,126]. The T2D management strategies using circadian intervention are shown in Figure 3.

## 4. Conclusions and Future Perspectives

Both animal models and human studies identify the inter-link between circadian arrhythmia and T2D. It is clear that T2D animals and human subjects experience significant circadian disruption, including irregular hormone secretion, the loss of sleep cycles, and irregular clock gene, protein, and metabolite expression in various metabolic tissues. The body’s homeostasis is maintained by the coordination of physiological actions across organs and tissues. On acellular level, glucose homeostasis maintained by the insulin hormone, which is produced by the pancreas, helps the utilization of blood glucose by metabolic tissues, such as muscle, fat, and the liver, where it is used for energy production. These metabolic tissues possess endogenous biological clocks and are involved in the regulation of glucose metabolism and ATP generation. The misalignment of endogenous biological clocks in metabolic tissues to impaired glucose homeostasis causes T2D [24,25,26]. Several altered metabolites have been discovered in clock-dysregulated T2D conditions that are associated with the gut microbiome [55]. However, more untargeted omics studies are necessary to determine how clock dysregulation in T2D is linked to the microbiota, as well as how the metabolic illness is impacted by the gut microbiota clock arrhythmicity. Further, circadian-dysregulated omics studies found several clock genes, proteins, and metabolites that were altered in the liver and skeletal muscles under chronic diet stress and under T2D conditions [32,61,62]. Additionally, global untargeted omics studies under various disease conditions are required to understand the mechanisms alterations in these tissues under the clock-dysregulated T2D state. Transcriptomics and metabolomics investigations found several altered genes and metabolites, notably lipid variants in pancreatic tissue, with clock-dysregulated T2D conditions. More proteomics studies are needed to explore the altered proteins in pancreatic tissues under clock-dysregulated T2D. A limited number of reports are available on omics studies of the heart and kidney under clock-dysregulated T2D conditions. Untargeted multi-omics studies need to be taken into consideration for the identification of perturbations in the genes, proteins, and metabolites in these tissues under clock-dysregulated conditions of T2D.

Integrating multi-omics approaches to investigate human physiology related to circadian rhythms and T2D on a cellular and tissue level will provide new opportunities for pharmacological interventions and disease management.

## Figures and Tables

**Figure 1 biology-12-00301-f001:**
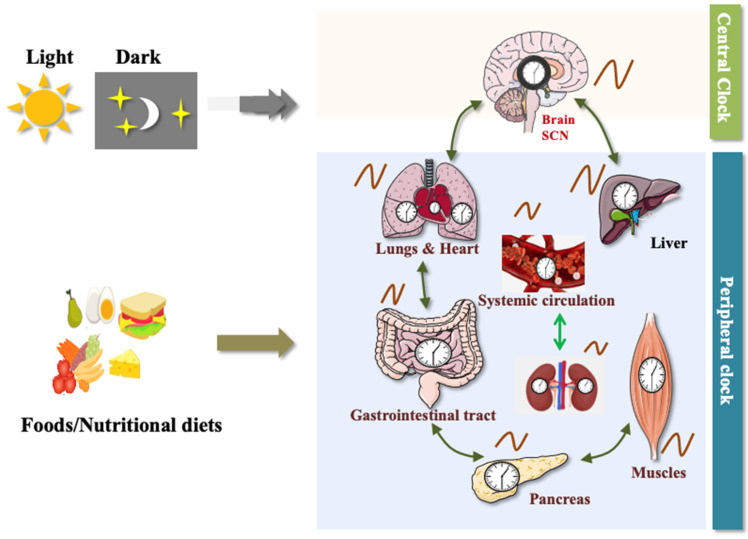
Typical circadian system of mammals: Central and peripheral clocks make up the mammalian circadian system. Peripheral clocks are controlled by food, but the central clock is controlled by photic signal. Suprachiasmatic nucleus (SCN) in the brain connects with peripheral clocks, such as the liver, heart, GI tract, and pancreas, as well as muscles, via synaptic and diffusible signals that regulate daily cycles of behavior and physiological activities. The peripheral tissues themselves have a clock and may interact with one another (figure source: Powerpoint & Adobe illustrator).

**Figure 3 biology-12-00301-f003:**
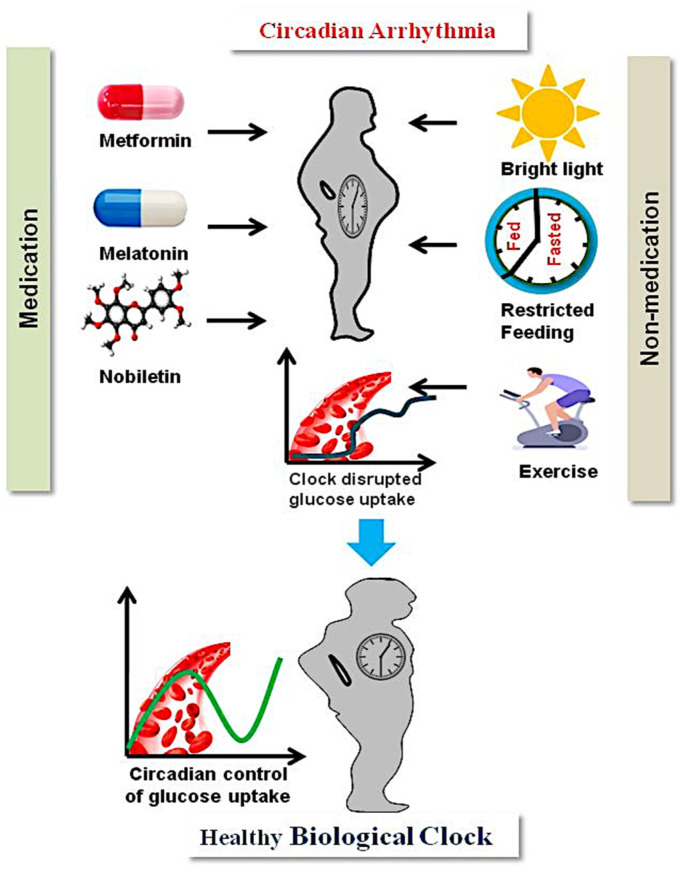
Management of type 2 diabetes through circadian intervention. Medication (metformin, melatonin, and nobiletin) and non-medication (bright light treatment, limited diet, and regular exercise) strategies may help to lessen the symptoms of clock dysregulated metabolic condition (T2D) (figure source: Powerpoint & Adobe illustrator).

**Table 1 biology-12-00301-t001:** Dysregulated clock genes and their effects in the heart *.

Dysregulated Clock	Effects
Cardiomyocyte-specific *Clock*-mutant mice	8% of the proteome exhibits a physiological variation.
Cardiomyocyte-specific *Bmal1*-KO mice	Heart metabolic abnormalities, dilated cardiomyopathy, and early mortality are all caused by the disruption of physiological variation in 10% of the transcriptome.
Specific vascular-smooth-muscle cell specific *Bmal1*-KO mice	24-h rhythm in blood pressure is distorted, and the moment of its peak has changed.
*Klf15*-KO mice	Greater vulnerability to ventricular arrhythmias and loss of physiological rhythm in ventricular repolarization length.
*Per2*-KO mice	Endothelial dysfunction
*Bmal1*-KO mice	Loss of physiological rhythms in heart rate and blood pressure
*Dbp−/−Hlf−/−Tef−/−* mice	Low levels of aldosterone, cardiomyopathy, cardiac hypertrophy, and left ventricular dysfunction.

* Note: (Ref. [83]).

**Table 2 biology-12-00301-t002:** Circadian clock-controlled transport genes in kidney *.

Gene	Functions	Models
Aqp2	Involved in water transport	CCD cell line
Aqp4	Involved in water transport	CCD cell line
Slc9a3 (NHE3)	Involved in Sodium/hydrogen exchange	Kidney
Gilz	Leucine zipper protein/regulation of sodium transport	DCT, CNT, CCD. Whole kidney
V1aR	Vasopressin receptor/regulation of water balance	DCT, CNT, CCD. Whole kidney
Slc6a9	Involved in glycine transport	DCT/CNT
Scnn1a (αENaC)	Alpha subunit of epithelial sodium channel	Cortex, outer medulla and inner medulla

* Note: (Ref. [99]).

**Table 3 biology-12-00301-t003:** Multi-omics studies for circadian dysfunction and insulin resistance (T2D).

Model System	Altered Circadian Rhythm Markers	Omics	References
Human (skeletal muscles)	*PER* and *CRY* blunted and PPAR gene enhanced in circadian-misaligned men.	Genomics	[66]
Human (skeletal muscles)	*BMAL1, CLOCK,* and *PER3*	Genomics	[67]
Male mice (BAT)	*Bmal1* and *Clock* showed altered expression and GmprImpdh1and Ucp1 also showed drastic alterations on HFD.	Genomics	[69]
Human	Study-I. *BMAL1, PER1, PER2, PER3* level reduced by more in T2D than ND Study-II. *BMAL1, PER1* and *PER3* reduced by more in T2D than ND.	Genomics	[69]
Human	*Per1-3, Cry2, Rev-erba, Clock,* and *Dbp* levels were lower in T2D compared to ND islet cells.	Genomics	[77]
Human	*CLOCK, BMAL1, PER1, CRY1*, and *CRY2* levels were reduced in T2D compared to ND in peripheral blood leucocytes.	Genomics	[28]
Human	*Per2, Per3*, and *Cry2* mRNA levels were reduced in T2D	Genomics	[78]
Mice	mRNA expression of Per2 and Bmal-1 significantly elevated	Genomics	[53]
Human (Plasma)	62 proteins of 1129 proteins altered, associated with multiple biological functions	Proteomics	[112]
KO mice (Cry1/2 & Bmal1)	Deregulated acetylation level of protein involved in TCA and urea cycle.	Proteomics	[60]
Human (skeletal muscles)	Lipoproteins were higher and mitochondrial complex III abundance was lower after morning HIT compared to after afternoon HIT.	Proteomics	[72]
Male mice	PUFA, diacylglycerols, phospholipids, sphingolipids, glycerolipids, and lysolipids and arginine, proline lysine, BCAA leucine, isoleucine & valine, and dipeptides altered on HFD	Metabolomics	[69]
Mice	Plasma glucose levels and hepatic glycogen disturbed by shift work	Metabolomics	[63]
Human pancreatic islets	Phosphatidylinositol (PI) and phosphatidylethanolamine [PE].	Metabolomics	[81]
Human (Plasma)	Amino acids, biogenic amines, acylcarnitine, sphingolipids significantly higher and metabolites, including glutamine, histidine, ornithine, serine, (octadecanoyl carnitine), octadecadienylcarnitine, lysoPC, PC, and 2Sphingolipids [SM C16:0 and SM (OH) C16:1], were significantly lower.	Metabolomics	[32]
Mice	Histidine and beta alanine showed a significant reduction and histamine and its derivatives increased in the diabetic mice.	Metabolomics	[53]
Human	Polyunsaturated triradylglycerol was abundant and phospholipid cardiolipin (CL) was low	Metabolomics	[80]
Human	Increased plasma diacylglycerols, skeletal muscle acyl-carnitines, and subcutaneous adipose tissue, and sphingomyelins and lysophospholipids.	Metabolomics	[72]

## Data Availability

The data presented in this study are available in this review article.

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
