# Peer review of "Multi-Omics Reveal Interplay between Circadian Dysfunction and Type2 Diabetes"

_biology, 2023, doi:10.3390/biology12020301_

Round 1
Reviewer 1 Report
The presented text thoroughly deals with a current and from a scientific and clinical point of view a very relevant topic. In terms of content, I think it is at a very good level. Some passages are quite complex, perhaps the clarity of the article would benefit from the insertion of additional images, diagrams or tables. The weaker side is the formal level of the text, e.g.:
- spaces between words or between a full stop and a new sentence are sometimes missing
- if your journal has no other usage, gene names should be written in italics
- capitalization of the names of human vs. of other organisms (e.g.: CLOCK and Clock are used for the human gene)
- two chapters are labeled 2.4
- chapter 3 is missing
etc.
I recommend that the article be accepted for publication after removing these deficiencies.
Author Response
Reviewer 1:
The presented text thoroughly deals with a current and from a scientific and clinical point of view a very relevant topic. In terms of content, I think it is at a very good level. Some passages are quite complex, perhaps the clarity of the article would benefit from the insertion of additional images, diagrams or tables. The weaker side is the formal level of the text, e.g.: I recommend that the article be accepted for publication after removing these deficiencies.
Response: Thank you for your positive response and your suggestions to improve the quality of the manuscript. All the suggestions have been implemented in the manuscript and the corresponding response to the comments are provided point wise below. Additional image figure 3 and tables, table 1 and table 2 were included as per the suggestion.
- spaces between words or between a full stop and a new sentence are sometimes missing
Response: As per your suggestions we have made the changes in manuscript
- if your journal has no other usage, gene names should be written in italics
- capitalization of the names of human vs. of other organisms (e.g.: CLOCK and Clock are used for the human gene)
Response: As per your suggestions we have made the changes (including genes names in italics and in capitals) in manuscript.
- two chapters are labeled 2.4
Response: It is corrected as per suggestion.
- chapter 3 is missing
Response: It is corrected as per suggestion
Reviewer 2 Report
Dear Authors,
It was a pleasure to read and review the extremely interesting review article “Multi-Omics reveal Interplay between Circadian dysfunction 2 and Type 2 Diabetes.” Indeed, multi-omics represent a major topic of interest nowadays.
Major Comments:
1. You could add another paragraph regarding cardiovascular system and circadian clock or/and multi-omics. An interesting study was published by Daios et al. (Medicina (Kaunas). 2021 Jan 6;57(1):41. doi: 10.3390/medicina57010041) regarding the effect of Circadian Pattern in patients with Acute Myocardial Infarction.
2. Given that the topic is extremely novel and interesting, you could add a “future perspectives section”.
3. You could also add a figure regarding the management of T2D through circadian intervention (section 5).
Minor Comments:
Minor English Editing is required
To Sum up,
The paper seems original with a considerable research interest in the recent literature in the field of Multi-Omics. It provides information and inspires thoughts in the researched area. It was a pleasure to read, and the readership is going to benefit.
Author Response
Dear Authors,
It was a pleasure to read and review the extremely interesting review article “Multi-Omics reveal Interplay between Circadian dysfunction 2 and Type 2 Diabetes.” Indeed, multi-omics represent a major topic of interest nowadays.
Response: Thank you for your positive response and your suggestions to improve the quality of the manuscript. All the suggestions have been implemented in the manuscript and the corresponding response to the comments are provided point wise below.
Major Comments:
- You could add another paragraph regarding cardiovascular system and circadian clock or/and multi-omics. An interesting study was published by Daios et al. (Medicina (Kaunas). 2021 Jan 6;57(1):41. doi: 10.3390/medicina57010041) regarding the effect of Circadian Pattern in patients with Acute Myocardial Infarction.
Response: Many thanks for your suggestion. A Paragraph regarding cardiovascular system and circadian clock is included in section 2.6 asper suggestion. The suggested reference was included in the manuscript at section 2.6 with reference number [90] as per suggestion.
- Given that the topic is extremely novel and interesting, you could add a “future perspectives section”.
Response: Many thanks for the suggestion. Future perspectives section was added and integrated with conclusion in section 4 (Conclusions and Future prospectives as per the suggestion.
- You could also add a figure regarding the management of T2D through circadian intervention (section 5).
Response: Many thanks for your suggestion. Figure 3 for management of Type 2 diabetes through circadian intervention was added as per the suggestion.
Minor Comments:
Minor English Editing is required
Response: English edits for complete manuscript was performed as per the suggestion.
To Sum up,
The paper seems original with a considerable research interest in the recent literature in the field of Multi-Omics. It provides information and inspires thoughts in the researched area. It was a pleasure to read, and the readership is going to benefit.
Response: Many thanks for your positive comments. We have implemented all the suggestions in the manuscript.
Reviewer 3 Report
The aim of this manuscript is to review the association between Circadian and Type 2 DM. The authors review the molecular mechanisms of circadian. They then discussed the role of circadian in different organs systems that related Type 2 DM by focusing on molecular mechanisms.
The overall review is interesting as they focus on the molecular mechanisms, signaling pathway and gene expression on this hot topic. However, the whole organization of this Review is confused, further revise needed before publication.
1. Sub-title on Line 149 need modified as the whole section reviewed the effect of circadian in different organs.
2. Where is section “3” ?
3. It would be better to integrate section “2” with section “4”, Focus on different organs, expended with further detail as Section “4” did. It will make more earlier for the reader.
4. The role of circadian in renal and cardiovascular systems should discussed.
5. It would be better to provide a molecular mechanism pathway in schematic figure in different organs.
Author Response
The aim of this manuscript is to review the association between Circadian and Type 2 DM. The authors review the molecular mechanisms of circadian. They then discussed the role of circadian in different organs systems that related Type 2 DM by focusing on molecular mechanisms.
The overall review is interesting as they focus on the molecular mechanisms, signaling pathway and gene expression on this hot topic. However, the whole organization of this Review is confused, further revise needed before publication.
Response: Thank you for your positive response and your suggestions to improve the quality of the manuscript. All the suggestions have been implemented in the manuscript and the corresponding response to the comments are provided point wise below.
- Sub-title on Line 149 need modified as the whole section reviewed the effect of circadian in different organs.
Response: Thank you for suggestion. We have integrated section 4 with section 2 (as per your comment 3), modified the heading for section 2 that incorporates various tissues in section.
- Where is section “3” ?
Response: Apologies for typo error, section 4 was supposed to be section 3. It was corrected in the revised manuscript as per suggestion.
- It would be better to integrate section “2” with section “4”, Focus on different organs, expended with further detail as Section “4” did. It will make more earlier for the reader.
Response: Section 2 (Circadian rhythms and T2D) and section 4 (Mutli-omics approaches in understanding circadian dysfunction in type 2diabetes) are integrated into single section i.e., section 2, where information about the circadian function of the organs and the omics studies in relation to Type 2 diabetes were discussed as per the suggestion.
- The role of circadian in renal and cardiovascular systems should discussed.
Response: Many thanks for your suggestion. The role of circadian in renal and cardiovascular systems are added in section 2.7 and section 2.6 respectively.
- It would be better to provide a molecular mechanism pathway in schematic figure in different organs.
Response: Many thanks for your suggestion. The core clock mechanism of all tissues/cells relies on Transcription and translation feedback loop (TTFL) and the additional non transcriptional mechanisms for sustaining the clock are discussed in section 1.1. Very limited (or no) information available in the literature for circadian dysregulated pathways in detail for many tissues including heart, renal, pancreas and liver etc in relation to Type 2 Diabetes. More number of research groups are started working to delineate the mechanisms for each tissue. In the present manuscript, we tried to incorporate the dysregulated genes, proteins and metabolites under Type 2 Diabetic conditions.
Round 2
Reviewer 3 Report
The authors response to all the comments with extensive modifications. The manuscript is more organized and easy for the reader as well.
This manuscript should be considered to published in this journal.